# Clinical efficacy of different methods for treatment of granulomatous lobular mastitis: A systematic review and network meta-analysis

**Yuxiang Zhou[1,2], Leilai Xu[3]***

**1** Zhejiang Chinese Medical University, Hangzhou, China, **2** The First People's Hospital of Yuhang District, Hangzhou, China, **3** First Affiliated Hospital of Zhejiang Chinese Medical University Hangzhou, Hangzhou, China

* xll1005@126.com

## Abstract

**Data Availability Statement:** All relevant data are within the manuscript and its Supporting Information files.

**Funding:** The author(s) received no specific funding for this work.

### Objective

This network meta-analysis aims to evaluate the recurrence rates of various treatment options for granulomatous lobular mastitis.

### Methods

We systematically searched and identified eligible studies in PubMed, EMBASE, Cochrane Library, and Web of Science databases until September 30, 2023. Original studies reporting the recurrence rates of various treatments were included. Subsequently, literature screening, data extraction, and network meta-analysis were conducted. This study was registered with PROSPERO (registration number CRD 42023434773).

### Results

Nineteen articles involving 1,095 patients were included in this study. The network meta-analysis revealed that several treatment combinations reduced the recurrence rate compared to observation: Surgery + Local steroid injection + Systemic steroids therapy (OR: 0.23, 95% CI 0.01 to 4.53), Local steroid injection (OR: 0.34, 95% CI 0.02 to 6.81), Surgery + Systemic steroids therapy (OR: 0.36, 95% CI 0.02 to 5.29), Surgery + Traditional Chinese Medicine (OR: 0.33, 95% CI 0.01 to 9.11), Systemic steroids therapy + MTX (OR: 0.62, 95% CI 0.01 to 34.59), and Systemic steroids therapy + drainage (OR: 0.76, 95% CI 0.05 to 10.67). Among these, Surgery + Local steroid injection + Systemic steroids therapy demonstrated superior efficacy. The surface under cumulative ranking curve (SUCRA) values were highest for Surgery + Local steroid injection + Systemic steroids therapy (0.85), followed by Local steroid injection (0.78) and Surgery + Systemic steroids therapy (0.77).

**Competing interests:** The authors have declared that no competing interests exist.

## Conclusions

Steroid-based combination therapy may be the first choice for IGM patients, with a comprehensive strategy of local and systemic steroids combined with surgery having the best effect on IGM.

## Introduction

Granulomatous mastitis, also known as Granulomatous lobular mastitis, is a benign chronic inflammatory condition of the breast tissue [1]. This condition was first described by Kessler and Wolloch in 1972. Granulomatous mastitis is characterized by the absence of an apparent underlying cause, making its diagnosis and management particularly challenging [2]. The condition predominantly affects women in their 30s and 40s, a demographic typically within the childbearing age range. It is most commonly observed within a few years following childbirth. It is particularly prevalent within a few years postpartum, highlighting a potential link with the physiological changes that occur during and after pregnancy. Despite its benign nature, the lack of a clear underlying cause makes granulomatous mastitis a condition that requires careful clinical attention to manage effectively [3].

Idiopathic granulomatous mastitis (IGM) is a rare, benign inflammatory condition characterized by noncaseating granulomatous inflammation within the breast lobules, with possible microabscess formation. The diagnosis of IGM requires a multifaceted approach, integrating clinical, radiological, or sonographic evidence with histological examination to confirm the presence of characteristic granulomas [4, 5]. The etiology and pathogenesis of IGM remain unclear, making it a challenging condition to understand and manage. However, several factors are suspected to contribute to its development. These include α1-antitrypsin deficiency, smoking, Corynebacterium infection, hyperprolactinemia, use of oral contraceptives, and autoimmune abnormalities. Despite its benign nature, the lack of a clear underlying cause necessitates careful clinical attention to effectively diagnose and manage the condition [6–8].

Idiopathic granulomatous mastitis (IGM) poses a challenge in clinical management due to the absence of a standardized treatment regimen. Treatment options are diverse, including antibiotics, steroids, immunosuppressive therapies like methotrexate, surgical interventions such as wide local excision or mastectomy, traditional Chinese medicine, and observation. Clinician experience plays a pivotal role in the selection of treatment modalities. Effectiveness and recurrence rates vary across studies, contributing to the complexity of decision-making [5, 9–12].

Meta-analyses have been limited by the lack of direct comparisons among interventions [13, 14]. To address this issue, we employed a network meta-analysis approach to directly and indirectly evaluate and compare various interventions. This method provides valuable insights into the efficacy of different treatment options, offering a more informed basis for clinical decision-making in the management of IGM.

## Materials and methods

### Search strategy

This network meta-analysis adhered to the PRISMA-NMA guidelines to ensure a rigorous and systematic review of healthcare interventions [15]. A comprehensive search strategy was implemented, utilizing MeSH terms and keywords related to treatment, management,

intervention, remission, recurrence, prognosis, idiopathic, mastitis, and granulomatous conditions. The search spanned several major databases, including PubMed, EMBASE, the Cochrane Library, and Web of Science, covering studies available up to September 30, 2023. Only English-language studies were included, focusing exclusively on human interventional studies. This methodical approach aimed to provide a thorough and precise synthesis of the current evidence base.

## Study selection criteria

The study selection process was meticulously carried out by two independent reviewers, FSY and XLF, with any disagreements resolved through thorough discussion.

Inclusion Criteria:

Study Types: Included were published research articles encompassing randomized controlled trials, cohort studies, and case/control studies.

Population: Studies involving human clinical trials with various interventions for Idiopathic Granulomatous Mastitis (IGM).

Interventions: The interventions considered included oral steroids, topical steroids, methotrexate (MTX), drainage, and surgical management such as lumpectomy, wide local excision, and mastectomy, as well as observation and prolactin-lowering agents, either alone or in combination.

Diagnosis: All included studies confirmed IGM diagnosis through histopathological examination.

Outcomes: Studies that reported recurrence rates of the different treatments for IGM.

Exclusion Criteria:

Publication Types: Conference abstracts, comments, letters, animal studies, reviews, case reports, ecological studies, cross-sectional studies, in vitro studies, and duplicate publications were excluded.

Data Reporting: Studies that did not report or calculate the desired parameters were excluded.

## Data extraction and quality assessment

In this meta-analysis, a meticulous approach was undertaken for quality assessment and data extraction to uphold the highest methodological standards. Two independent investigators, ZYX and XLL, conducted these processes to ensure objectivity and minimize bias. The data extraction checklist included critical elements such as the first author, publication year, patient number, recurrence rate, intervention details, sample size, follow-up period, and observation indices.

Quality assessment was performed using the Newcastle–Ottawa Quality Assessment Scale (NOS), a validated tool for evaluating cohort and case-control studies. The NOS criteria encompass patient selection, group comparability, and ascertainment of exposure and outcomes. Studies were classified according to their NOS scores: 7–8 points indicated very good quality, 5–6 points denoted good quality, 4 points were satisfactory, and scores of 0–3 points were considered unsatisfactory.

## Data analysis

A network meta-analysis was conducted using odds ratio (OR) values to evaluate the recurrence rate of idiopathic granulomatous mastitis (IGM), with a 95% confidence interval (CI) for both indirect and mixed comparisons. The assessment included checks for similarity, transitivity, and consistency.

For similarity, a qualitative assessment was conducted on the clinical and methodological characteristics of the included studies, examining patient populations, interventions, and outcomes to ensure comparability and meaningfulness in comparisons. Transitivity was evaluated by analyzing the distribution of relevant covariates across different treatment groups, confirming no significant disparities that could compromise comparability [16]. Consistency was formally assessed through a combination of global and local inconsistency evaluations. Global and local inconsistency were examined through a $\chi^2$ test and side-splitting, respectively. Probability estimates for ranking each intervention were calculated, and comparison-adjusted funnel plots were used to evaluate publication bias. For studies with missing data, attempts were made to contact the original authors to retrieve the missing information. When this was not feasible, analyses were based on the available data (complete case analysis). Imputation methods, such as mean imputation for missing continuous variables, were applied where appropriate. Sensitivity analyses were conducted to evaluate the potential impact of missing data, and all methods used were carefully documented. Additionally, sensitivity analyses were carried out to assess the influence of individual studies, excluding those with high risk of bias or those contributing to global or local inconsistency. Statistical evaluations of inconsistency and the creation of network graphs and figures were carried out using the network and network graphs packages in STATA version 15 (STATA Corporation, College Station, TX, USA). Following preliminary searches, the protocol was registered with PROSPERO (registration number CRD42023434773). All analyses were grounded in previously published studies, obviating the need for ethical approval and patient consent.

## Results

### Search results

The initial search of the databases yielded 573 articles. Following the elimination of duplicates, 376 papers underwent title and abstract screening, resulting in the exclusion of 329 papers. Subsequently, 47 papers were assessed for eligibility through full-text review, with 19 studies ultimately included in the meta-analysis [17–35]. The PRISMA flow diagram illustrating the study selection process is presented in **Fig 1.**

### Eligible studies and patient characteristics

The basic characteristics of the eligible studies are outlined in Table 1. All studies included were published in English between 2013 and 2022. No randomized controlled trials were identified; the majority of the studies were retrospective. The most frequently employed interventions for IGM were steroids, surgical management, and a combination of steroids and surgical management. Most of the studies included had two arms, with only two studies having three or four arms. The risk of bias assessment for the 19 included trials are summarized in **Table 1**, evaluated using the Newcastle-Ottawa Scale (NOS) checklist. The median NOS quality score for cohort studies ranged from 6 to 7, indicating medium to high quality.

### Network diagrams

The 19 included studies covered thirteen different treatments: Observation(Obs), Systemic steroids therapy(SST), Local steroid injection(LSI), Surgery(Sur), Surgery+ Systemic steroids therapy(SurSST), Systemic steroids therapy+drainage (SSTDra), drainage(Dra), Surgery+ Traditional Chinese Medicine(SurTCM), Surgery+Local steroid injection+Systemic steroids therapy (LSISSTSur), Systemic steroids therapy+ Antibiotics(SSTATB), Systemic steroids therapy +MTX(SSTMTX), Antibiotics(ATB), Antibiotics+drainage(ATBDra). The maximum sample

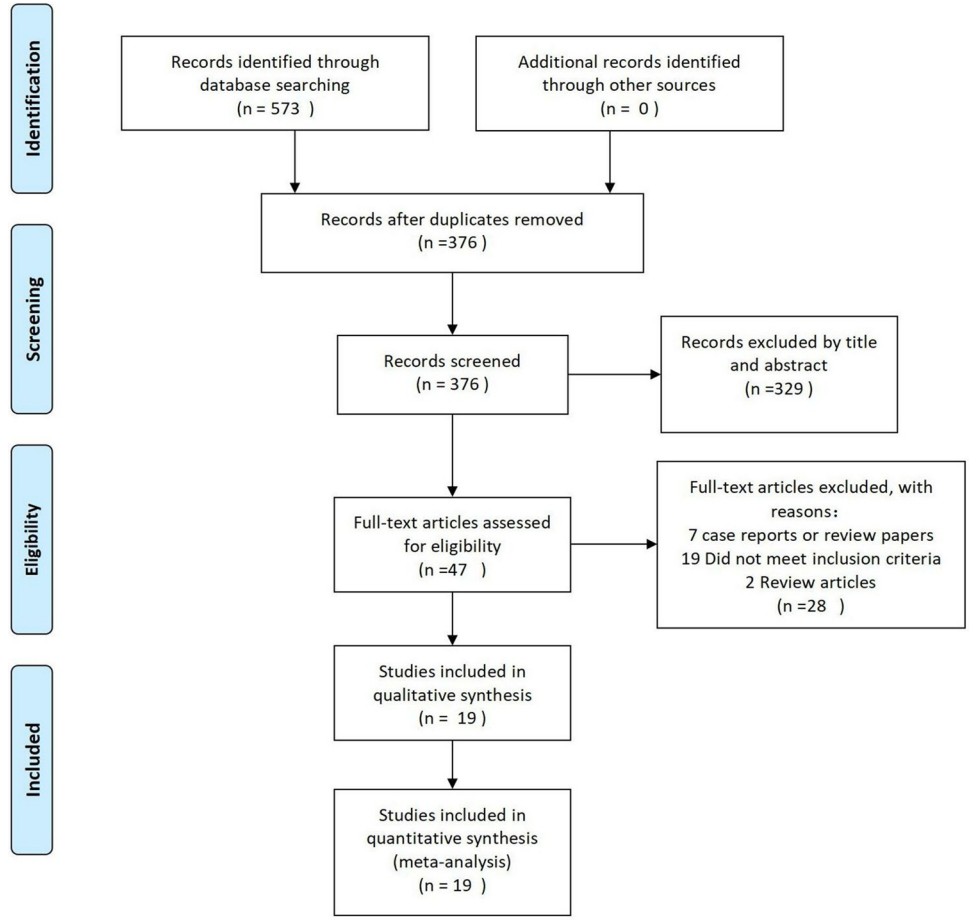

**Fig 1. Flowchart of the study.**

size of treatment was Systemic steroids therapy and Surgery, shown in **Fig 2**. The thickness of the lines in the network graph is proportional to the number of studies included in each pairwise comparison, and the diameter of the circles is proportional to the number of participants who received each intervention.

## Testing for inconsistency

The p-value for the test of overall inconsistency was greater than 0.05 (p = 0.9682). Similarly, no p-values were below 0.05 for the test of local inconsistency. Neither the global nor the local tests showed significance, indicating that the assumption of consistency was upheld.

## Outcomes of the network meta-analysis

Network meta-analysis showed that, in comparison with observation, three-combined treatment (Surgery+Local steroid injection+ Systemic steroids therapy) ranked the best for the recurrence rate of different treatments for IGM reduced(OR: 0.23, 95%CI 0.01 to 4.53), followed by Local steroid injection(OR: 0.34, 95%CI 0.02 to 6.81), two-combined treatment (Surgery+Systemic steroids therapy)(OR: 0.36, 95%CI 0.02 to 5.29), (Surgery+ Traditional Chinese Medicine)(OR: 0.33, 95%CI 0.01 to 9.11), (Systemic steroids therapy+ MTX)(OR: 0.62, 95%CI 0.01 to 34.59), (Systemic steroids therapy+ drainage)(OR: 0.76, 95%CI 0.05 to 10.67), all of

**Table 1. Characteristics of included studies.**

| Study | Intervention | Sample size | Total recurrence | Recurrence rate (%) | Follow-up | Quality assessment | Type of study |
|---|---|---|---|---|---|---|---|
| Fatih 2022 [35] | Systemic steroids therapy | 16 | 2 | 12.5 | 1–5 months | 8/9 | Cohort |
| | Local steroid injection | 42 | 2 | 4.8 | 1-12months | | |
| Jiang 2019 [34] | Systemic steroids therapy | 44 | 10 | 22.7 | 12–36 months | 8/9 | Cohort |
| | Surgery+Systemic steroids therapy | 156 | 8 | 5.1 | | | |
| Taha 2021 [33] | Local steroid injection | 38 | 0 | 0 | 12 months | 7/9 | Cohort |
| | Surgery | 48 | 15 | 31.3 | | | |
| Hakan 2015 [32] | Systemic steroids therapy | 44 | 9 | 20.5 | 1–38 months | 8/9 | Cohort |
| | Surgery | 33 | 0 | 0 | 1–120 months | | |
| Kazuhisa 2013 [17] | Systemic steroids therapy | 10 | 0 | 0 | 6–104 months | 7/9 | Cohort |
| | Surgery | 2 | 2 | 100 | 26–77 months | | |
| Prakasit2018 [31] | Systemic steroids therapy | 6 | 1 | 16.7 | 5–12 months | 7/9 | Cohort |
| | Surgery | 30 | 4 | 13.3 | | | |
| Leyla 2021 [30] | Systemic steroids therapy | 23 | 9 | 39.1 | 8–48 months | 7/9 | Cohort |
| | Surgery | 17 | 7 | 41.2 | | | |
| | Systemic steroids therapy +drainage | 47 | 9 | 19.1 | | | |
| Alper 2014 [29] | Surgery | 53 | 4 | 7.5 | 3–170 months | 6/9 | Cohort |
| | Surgery+Systemic steroids therapy | 21 | 0 | 0 | | | |
| Sung 2013 [28] | Observation | 8 | 0 | 0 | 14–40 months | 6/9 | Cohort |
| | Systemic steroids therapy | 13 | 1 | 7.7 | | | |
| | Surgery | 23 | 2 | 8.7 | | | |
| | Drainage | 14 | 1 | 7.1 | | | |
| Hasan 2014 [27] | Systemic steroids therapy | 23 | 7 | 30.4 | 2–18 months | 7/9 | Cohort |
| | Surgery+Systemic steroids therapy | 37 | 0 | 0 | 22–78 months | | |
| Lai 2005 [18] | Observation | 4 | 0 | 0 | 1–24 months | 6/9 | Cohort |
| | Surgery | 1 | 0 | 0 | 1–30 months | | |
| Liu 2020 [26] | Surgery | 50 | 1 | 2 | 1–12 months | 7/9 | Cohort |
| | Surgery+Traditional Chinese Medicine | 60 | 0 | 0 | | | |
| Zhang2020 [25] | Surgery | 25 | 4 | 16 | 3–23 months | 7/9 | Cohort |
| | Surgery+Traditional Chinese Medicine | 28 | 0 | 0 | 2–22 months | | |
| Ren 2013 [24] | Surgery+Systemic steroids therapy | 28 | 6 | 21.4 | 24 months | 7/9 | Cohort |
| | Surgery+Local steroid injection +Systemic steroids therapy | 34 | 5 | 14.7 | | | |
| Oran2013 [23] | Systemic steroids therapy | 25 | 5 | 20 | 3–135 months | 7/9 | Cohort |
| | Surgery | 18 | 3 | 16.7 | | | |
| SAKURAI 2011 [22] | Systemic steroids therapy | 5 | 0 | 0 | 1–39 months | 6/9 | Cohort |
| | Systemic steroids therapy +Antibiotics | 2 | 0 | 0 | | | |
| Sheybani 2015 [21] | Systemic steroids therapy | 15 | 3 | 20 | 6–22 months | 7/9 | Cohort |
| | Systemic steroids therapy+MTX | 6 | 0 | 0 | | | |
| Shin2017 [20] | Surgery | 20 | 5 | 25 | 22–98 months | 8/9 | Cohort |
| | Systemic steroids therapy +drainage | 14 | 1 | 7.1 | | | |

(*Continued*)

**Table 1.** (Continued)

| Study | Intervention | Sample size | Total recurrence | Recurrence rate (%) | Follow-up | Quality assessment | Type of study |
|---|---|---|---|---|---|---|---|
| Skandarajah2014 [19] | Surgery | 2 | 0 | 0 | 0–24 monhs | 6/9 | Cohort |
| | Antibiotics | 7 | 3 | 42.9 | | | |
| | Antibiotics+drainage | 3 | 2 | 66.7 | | | |

which ranked higher than observation(**Fig 3, Tables 2 and 3**). However, drainage, systemic steroids therapy, surgery, systemic steroids therapy+ antibiotics, and antibiotics, antibiotics+drainage, were ranked lower than observation. This network meta-analysis suggested that three combined treatments (Surgery+Local steroid injection+ Systemic steroids therapy) have the highest probability of reducing the recurrence rate of IGM. The surface under cumulative ranking curve (SUCRA) values for (Surgery+Local steroid injection+ Systemic steroids therapy) were 0.85, followed by Local steroid injection (SUCRA, 0.78) and (Surgery+Systemic steroids therapy) (SUCRA, 0.77).

## Publication bias

The comparison-adjusted funnel plots did not exhibit any signs of apparent asymmetry (Fig 4). Consequently, no significant publication bias was detected. No missing data were reported for the included studies. As a result, no imputation or data removal was necessary.

## Discussion

Several interventions effectively reduce the recurrence rate of idiopathic granulomatous mastitis (IGM). Meta-analyses by Lei et al. and Godazandeh et al. [13] indicate that combined

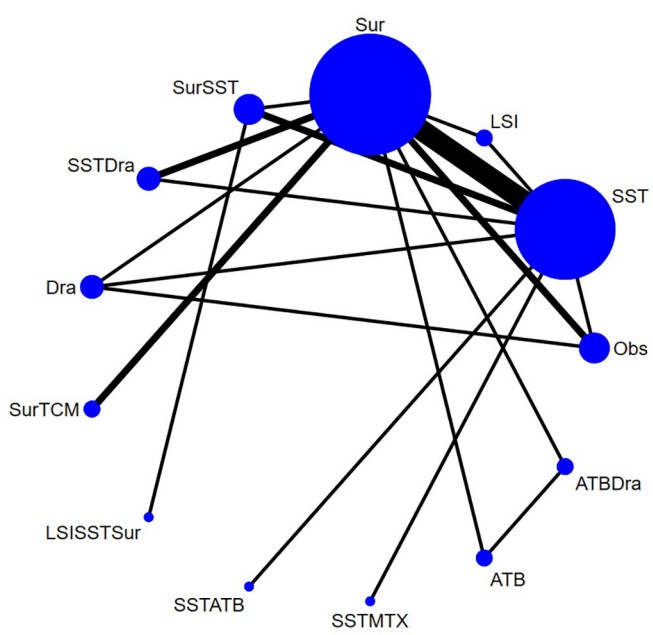

**Fig 2. Network plot of Granulomatous lobular mastitis in various treatments.**

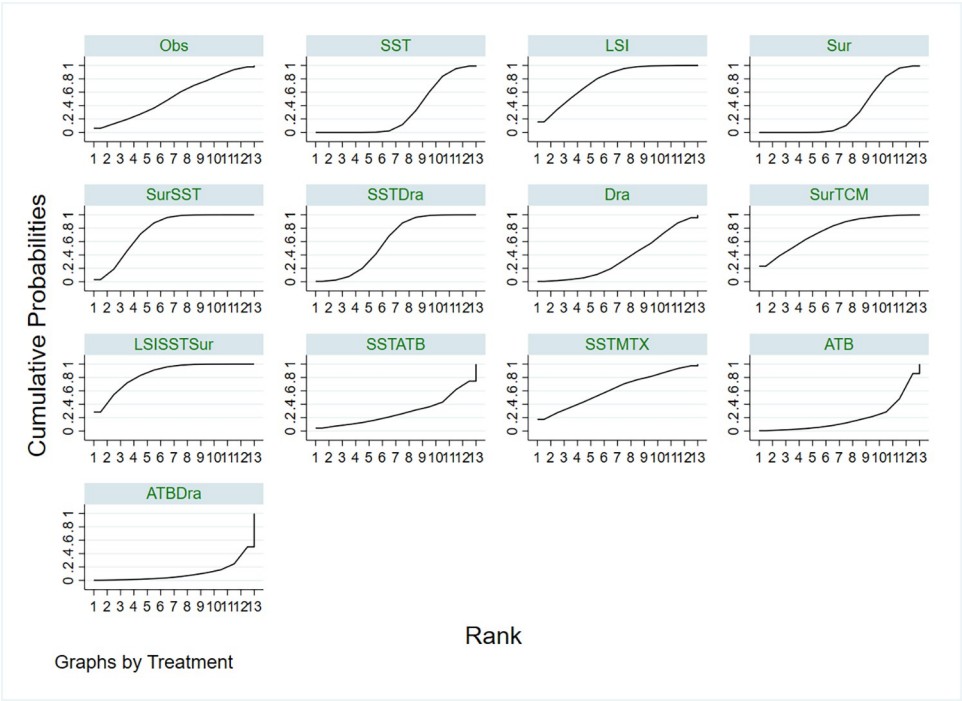

**Fig 3. Rankograms for the network show the probability of the recurrence rate of different treatments for IGM reduced.** Observation, Obs; Systemic steroids therapy, SST; Local steroid injection, LSI; Surgery, Sur; Surgery +Systemic steroids therapy, SurSST; Systemic steroids therapy+ drainage, SSTDra; drainage, Dra; Surgery+ Traditional Chinese Medicine, SurTCM; Surgery+Local steroid injection+ Systemic steroids therapy, LSISSTSur; Systemic steroids therapy+Antibiotics, SSTATB; Systemic steroids therapy+ MTX, SSTMTX; Antibiotics, ATB; Antibiotics+drainage, ATBDra.

surgical and corticosteroid treatment significantly lower the recurrence rate of IGM compared to corticosteroid or surgical monotherapy. Lei et al.'s meta-analysis, covering 15 studies, reported recurrence rates of 6.8%, 20.9%, and 4% for surgery alone, oral corticosteroids alone, and combined surgery and corticosteroids, respectively, highlighting the superior disease control offered by the combined approach. Similarly, Godazandeh et al. found lower recurrence rates with combined treatment than with corticosteroid or surgery alone, supporting this conclusion. The present study further validates the efficacy of combined surgery and corticosteroid therapy in reducing IGM recurrence.

Compared to previous research, this study incorporates post-2017 studies and employs network meta-analysis to advance the evaluation of relative treatment efficacies. Network meta-analysis, by integrating direct and indirect evidence, enhances comparability between different treatment regimens and ranks multiple treatment combinations using SUCRA values, providing more detailed information for treatment selection. This study reveals that triple therapy (surgery + local corticosteroid injection + systemic corticosteroids) performs best in recurrence control, followed by local corticosteroid injection combined with systemic corticosteroids and surgery. These findings suggest that an integrated regimen of local and systemic corticosteroids with surgery may be the optimal strategy to reduce IGM recurrence risk.

Since Lei et al.'s initial meta-analysis in 2017, corticosteroid application methods have improved [36, 37]. The latest studies in this analysis indicate that combining local corticosteroid injections with systemic administration not only acts directly on the lesion but may also reduce the risk of systemic drug side effects. This approach is particularly suitable for patients

**Table 2. Results of the recurrence rate of different treatments for IGM reduced; results presented as constant odds ratios between all competing interventions with 95% confidence intervals.**

Observation, Obs; Systemic steroids therapy, SST; Local steroid injection, LSI; Surgery, Sur; Surgery+Systemic steroids therapy, SurSST; Systemic steroids therapy+ drainage, SSTDra; drainage, Dra; Surgery+ Traditional Chinese Medicine, SurTCM; Surgery+Local steroid injection+ Systemic steroids therapy, LSISSTSur; Systemic steroids therapy + MTX, SSTMTX; Antibiotics, ATB; Antibiotics+drainage, ATBDra.

| LSISSTSur | | | | | | | | | | | | |
|---|---|---|---|---|---|---|---|---|---|---|---|---|
| 0.68 (0.07,6.82) | LSI | | | | | | | | | | | |
| 0.63 (0.17,2.34) | 0.93 (0.14,6.27) | SurSST | | | | | | | | | | |
| 0.69 (0.04,11.14) | 1.02 (0.06,16.57) | 1.09 (0.09,12.71) | SurTCM | | | | | | | | | |
| 0.37 (0.01,12.07) | 0.54 (0.02,18.64) | 0.58 (0.02,14.80) | 0.53 (0.01,25.65) | SSTMTX | | | | | | | | |
| 0.30 (0.05,1.91) | 0.44 (0.07,2.95) | 0.48 (0.13,1.75) | 0.44 (0.04,4.75) | 0.82 (0.03,21.29) | SSTDra | | | | | | | |
| 0.23 (0.01,4.53) | 0.34 (0.02,6.81) | 0.36 (0.02,5.29) | 0.33 (0.01,9.11) | 0.62 (0.01,34.59) | 0.76 (0.05,10.67) | Obs | | | | | | |
| 0.11 (0.01,1.39) | 0.16 (0.01,2.11) | 0.17 (0.02,1.54) | 0.16 (0.01,2.99) | 0.29 (0.01,11.98) | 0.36 (0.04,3.10) | 0.47 (0.03,8.88) | Dra | | | | | |
| 0.10 (0.02,0.49) | 0.15 (0.03,0.80) | 0.16 (0.06,0.39) | 0.15 (0.01,1.47) | 0.27 (0.01,6.17) | 0.34 (0.13,0.88) | 0.44 (0.03,5.65) | 0.94 (0.12,7.12) | SST | | | | |
| 0.10 (0.02,0.56) | 0.15 (0.03,0.83) | 0.16 (0.05,0.48) | 0.14 (0.02,1.28) | 0.27 (0.01,6.62) | 0.33 (0.13,0.86) | 0.44 (0.04,5.27) | 0.92 (0.13,6.74) | 0.98 (0.47,2.06) | Sur | | | |
| 0.05 (0.00,4.07) | 0.07 (0.00,6.22) | 0.07 (0.00,5.29) | 0.07 (0.00,8.00) | 0.12 (0.00,23.16) | 0.15 (0.00,11.29) | 0.20 (0.00,27.23) | 0.43 (0.00,45.03) | 0.45 (0.01,30.17) | 0.46 (0.01,32.69) | SSTATB | | |
| 0.03 (0.00,1.10) | 0.04 (0.00,1.63) | 0.04 (0.00,1.37) | 0.04 (0.00,2.01) | 0.07 (0.00,7.10) | 0.08 (0.00,2.75) | 0.11 (0.00,7.25) | 0.24 (0.00,11.61) | 0.25 (0.01,7.77) | 0.26 (0.01,7.27) | 0.56 (0.00,125.17) | ATB | |
| 0.01 (0.00,0.67) | 0.02 (0.00,1.00) | 0.02 (0.00,0.86) | 0.02 (0.00,1.22) | 0.03 (0.00,4.16) | 0.04 (0.00,1.72) | 0.05 (0.00,4.34) | 0.11 (0.00,7.08) | 0.12 (0.00,4.89) | 0.12 (0.00,4.61) | 0.26 (0.00,70.97) | 0.47 (0.04,5.46) | ATBDra |

Table 3. Treatment relative ranking.

| Treatment | SUCRA | PrBest | MeanRank |
|-----------|-------|--------|----------|
| LSISSTSur | 85.2 | 28.3 | 2.8 |
| LSI | 77.5 | 15.6 | 3.7 |
| SurSST | 76.9 | 3.1 | 3.8 |
| SurTCM | 75.9 | 23.2 | 3.9 |
| SSTMTX | 62.2 | 17.4 | 5.5 |
| SSTDra | 60.3 | 0.6 | 5.8 |
| Obs | 53.1 | 6.1 | 6.6 |
| Dra | 36.2 | 0.6 | 8.7 |
| SST | 32.1 | 0 | 9.1 |
| Sur | 31.7 | 0 | 9.2 |
| SSTATB | 28.8 | 4.4 | 9.5 |
| ATB | 19.6 | 0.5 | 10.6 |
| ATBDra | 10.5 | 0.2 | 11.7 |

with more extensive or profound lesions, as it enables more precise targeting of substantive lesions within breast tissue [38]. In contrast, systemic corticosteroid monotherapy has shown relatively limited efficacy in reducing recurrence rates.

The literature included in Lei [14], Godazandeh [13], and this study has some overlap, with certain limitations in these studies, including small sample sizes, variability in corticosteroid

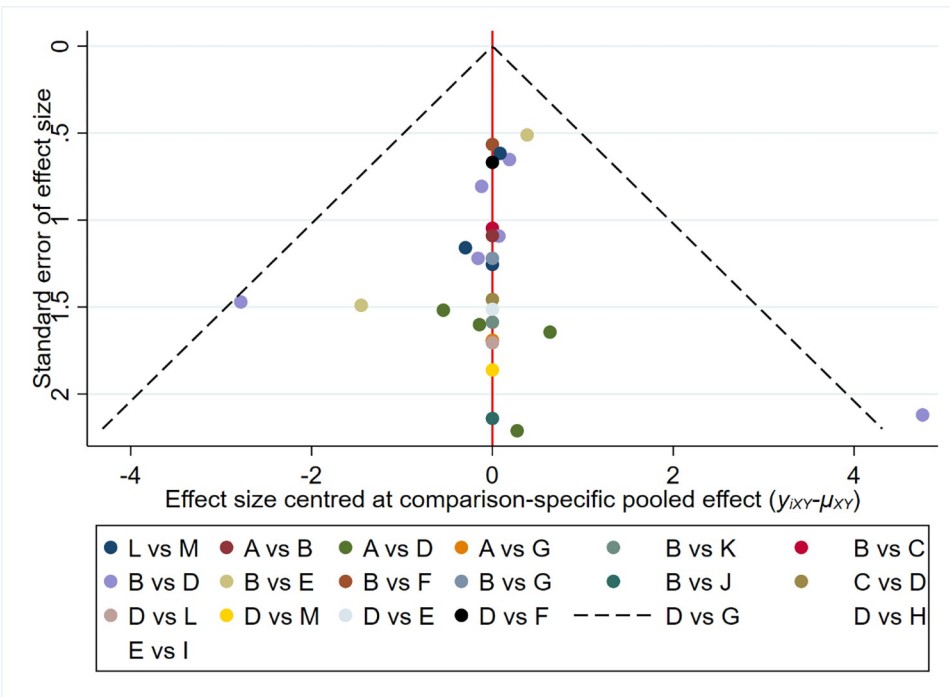

Fig 4. Comparison-adjusted funnel plot for the selected studies with the recurrence rate of different treatments for IGM. Abbreviation: A, Observation; B, Systemic steroids therapy; C, Local steroid injection;D, Surgery; E, Surgery +Systemic steroids therapy; F, Systemic steroids therapy+ drainage; G, Drainage; H, Surgery+Traditional Chinese Medicine; I, Surgery+Local steroid injection+ Systemic steroids therapy; J, Systemic steroids therapy+Antibiotics; K, Systemic steroids therapy+MTX; L, Antibiotics; M, Antibiotics+drainage.

dosing, and differing follow-up durations. By synthesizing the latest literature, our analysis builds on these previous studies, reinforcing the value of combination therapy and suggesting potential directions for optimizing treatment. Future research should aim to standardize dosing and assess long-term patient outcomes in larger-scale randomized studies to validate these preliminary findings.

Akbulut et al. [39] discovered that methotrexate could effectively prevent complications, alleviate the inflammatory process, and reduce the use of hormones through a retrospective analysis of 541 IGM patients. Early immunosuppressive treatment for IGM patients with infection and abscess formation can effectively shorten the course of the disease and reduce recurrence [3, 28]. Kim et al. [9] found that patients who used a combination of methotrexate and steroids were less likely to experience relapse and could maintain remission after discontinuing the medication. It can be concluded that the combined use of hormones and methotrexate may represent a reasonable and effective option for some patients with IGM. It should be noted that the dosage and course of treatment of methotrexate have not been standardized. Simultaneously, folic acid should be taken in combination to prevent folate deficiency syndrome, and attention should also be given to interstitial pneumonia caused by methotrexate. A rigorous evaluation is necessary before use [39].

Drainage requires an extended treatment duration, and the incision is challenging to heal, potentially resulting in the formation of fistulas. Consequently, the utilization of this treatment is a matter of controversy. Erozgen et al. [40] documented 13 patients with IGM and abscesses who underwent incision and drainage. Only 4 of them attained complete relief through drainage; however, the recovery time was prolonged. In the remaining 9 cases, lesions persisted after drainage, necessitating additional steroid therapy. Incision and drainage are options for treating IGM patients with abscesses; however, it is recommended to combine drainage with other therapeutic approaches.

This study revealed that the recurrence rate associated with observation was surpassed only by combination therapy involving steroids. Some cases of IGM are regarded as self-limiting [41]. Various drug treatment approaches documented in the literature necessitate a treatment duration of at least 6 to 9 months, aligning with the inherent self-healing timeline of the disease. Consequently, discerning whether this reflects the drug's efficacy or the natural course of the disease is challenging. Studies employing close observation and follow-up as the sole intervention reveal that most patients exhibit no evident clinical symptoms. Therefore, close monitoring is an optional nonsurgical treatment for patients diagnosed with GLM at an early stage, exhibiting mild symptoms, who prefer conservative treatment [18].

Given the typical clinical manifestation of IGM as breast inflammation, the majority of patients undergo early antibiotic treatment. However, a significant proportion of IGM patients lack microbiological evidence of bacterial infection, leading to frequent antibiotic treatment failures. Although some researchers have suggested that there may be a link between IGM and Corynebacterium infection, it has not been clear whether the positive result is caused by colonization or primary or secondary infection. Currently, no effective treatment for Corynebacterium exists. Therefore, the existing evidence supporting antibiotic treatment for IGM remains insufficient, and its efficacy is limited [42].

Traditional Chinese medicine is also considered a viable treatment option. The theory of syndrome differentiation and treatment allows for the adjustment of medications based on the specific symptoms exhibited by the patient. Comprehensive traditional Chinese medicine therapy is associated with a low recurrence rate, minimal tissue damage, and negligible impact on breast appearance [43].

This study has several limitations. Firstly, it did not evaluate the side effects of various interventions due to a lack of corresponding data. Secondly, the included studies had a

retrospective design, which introduces potential selection bias. Other limitations include the small sample sizes of some studies and the dosage variations of the same treatment regimen across different centers.

## Conclusions

The results of this network meta-analysis suggest that steroid-based combination therapy may be the primary choice for patients with idiopathic granulomatous mastitis (IGM). The combined treatment approach of local and systemic corticosteroids, along with surgery, may represent the most favorable therapeutic strategy for IGM. To support this perspective, additional large-scale randomized controlled trials should be conducted to provide a higher level of evidence.

## Supporting information

**S1 Checklist. PRISMA 2020 checklist.**
(DOCX)

**S1 Table. Table studies identified in the literature search.**
(DOCX)

**S2 Table. Table data extracted from the primary research sources.**
(DOCX)

**S3 Table. Table (NOS) checklist.**
(DOCX)

## Author Contributions

**Writing – original draft:** Yuxiang Zhou.

**Writing – review & editing:** Leilai Xu.

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
