## [Decision Letter · Decision Letter 0]

29 Sep 2024

PONE-D-24-22261Clinical efficacy of different methods for treatment of granulomatous lobular mastitis: a systematic review and network meta-analysisPLOS ONE

Dear Dr. XU,

Thank you for submitting your manuscript to PLOS ONE. After careful consideration, we feel that it has merit but does not fully meet PLOS ONE’s publication criteria as it currently stands. Therefore, we invite you to submit a revised version of the manuscript that addresses the points raised during the review process.

We look forward to receiving your revised manuscript.

Kind regards,

Serkan Yılmaz

Academic Editor

PLOS ONE

Journal Requirements:

5. As required by our policy on Data Availability, please ensure your manuscript or supplementary information includes the following: 

Additional Editor Comments:

POP

Reviewers' comments:

Reviewer's Responses to Questions

**Comments to the Author**

1. Is the manuscript technically sound, and do the data support the conclusions?

Reviewer #1: Partly

Reviewer #2: Partly

2. Has the statistical analysis been performed appropriately and rigorously? 

Reviewer #1: No

Reviewer #2: Yes

3. Have the authors made all data underlying the findings in their manuscript fully available?

Reviewer #1: No

Reviewer #2: Yes

4. Is the manuscript presented in an intelligible fashion and written in standard English?

Reviewer #1: Yes

Reviewer #2: Yes

5. Review Comments to the Author

Reviewer #1: In this work, the authors applied a meta-network analysis to evaluate treatment options for idiopathic granulomatous mastitis (IGM). The final network includes data from 19 articles. Odds ratio values were used to evaluate the recurrence rate of IGM, finding that the combination of three treatments: surgery, local steroid injection, and systemic steroids therapy, had the highest probability of reducing the recurrence rate. The authors followed a standard pipeline used in network meta-analysis studies to derive their results, and the study selection criteria are well described. However, there are some issues that need to be addressed:

Table 1 includes characteristics of the studies analyzed. However, there are at least nine articles that were not included in the References section. All studies used for this analysis must be added to the References list.

The authors should include a table with individual statistics for each study, such as the recurrence rate with the associated confidence intervals (CIs).

The Data Analysis section states, “The assessment included checks for similarity, transitivity, and consistency.” However, there is no description of the methods used for these checks.

The PROSPERO entry mentioned in the manuscript has a description of the Higgins I squared statistic thresholds used to assess heterogeneity (or inconsistency). However, these are not mentioned in the manuscript. A thorough description of all the methods used to derive the results is needed in the Data Analysis section.

The authors mention Lei’s meta-analysis study from 2017, which included 15 publications. The overlap between the selected studies should be mentioned, and a brief discussion of the improvements made in treatment options from that date until now should be included.

There is another publication that should be used to compare and discuss the results obtained from this analysis: Godazandeh, G., Shojaee, L., Alizadeh-Navaei, R., et al. "Corticosteroids in idiopathic granulomatous mastitis: a systematic review and meta-analysis." Surg Today 51, 1897–1905 (2021). These authors also found that the combination of surgery plus steroids has the lowest recurrence rate. Just as in the case for Lei, 2017, adding a brief discussion about the overlap and differences in the results could inform the reader about the diversity in conclusions made with similar methodologies.

Reviewer #2: Please see comments. The article is well written and the analysis is well documented. There seems one methodology error though pertaining to the use of Odds ratio instead of risk ratio, a mistake popularly done in the literature.

6. PLOS authors have the option to publish the peer review history of their article (what does this mean?). If published, this will include your full peer review and any attached files.

Reviewer #1: No

Reviewer #2: **Yes: **Muhammad Abdul Qadeer

---

## [Author Response · Author response to Decision Letter 0]

9 Dec 2024

Review Comments to the Author

Reviewer #1: In this work, the authors applied a meta-network analysis to evaluate treatment options for idiopathic granulomatous mastitis (IGM). The final network includes data from 19 articles. Odds ratio values were used to evaluate the recurrence rate of IGM, finding that the combination of three treatments: surgery, local steroid injection, and systemic steroids therapy, had the highest probability of reducing the recurrence rate. The authors followed a standard pipeline used in network meta-analysis studies to derive their results, and the study selection criteria are well described. However, there are some issues that need to be addressed:

（1）Table 1 includes characteristics of the studies analyzed. However, there are at least nine articles that were not included in the References section. All studies used for this analysis must be added to the References list.

Response： Thank you so much for your suggestions! We have added citations to the references in each study in Table 1.

（2）The authors should include a table with individual statistics for each study, such as the recurrence rate with the associated confidence intervals (CIs).

Response： Thank you so much for your suggestions! We have included the recurrence rate for each study, but unfortunately, we did not obtain the associated confidence intervals (CIs).

（3）The Data Analysis section states, "The assessment included checks for similarity, transitivity, and consistency." However, there is no description of the methods used for these checks.

Response： Thank you so much for your suggestions! Per the reviewer's requirements, the similarity, transitivity, and consistency methods have been described more clearly.

（4）The PROSPERO entry mentioned in the manuscript has a description of the Higgins I squared statistic thresholds used to assess heterogeneity (or inconsistency). However, these are not mentioned in the manuscript. A thorough description of all the methods used to derive the results is needed in the Data Analysis section.

Response： Thank you so much for your suggestions! We acknowledge that your suggestion is necessary. In network meta-analysis (NMA), the Higgins I² statistic helps assess heterogeneity and consistency of results. However, in NMA, we consider assessing consistency more critical than determining traditional heterogeneity, as consistency reflects the difference between the results of direct and indirect comparisons. NMA typically uses global consistency tests (such as the χ² test) and local consistency tests (such as side-splitting), which may directly assess the consistency issue within the network structure more than the I² does. Thank you once again for your valuable insights!

（5）The authors mention Lei's meta-analysis study from 2017, which included 15 publications. The overlap between the selected studies should be mentioned, and a brief discussion of the improvements made in treatment options from that date until now should be included.

There is another publication that should be used to compare and discuss the results obtained from this analysis: Godazandeh, G., Shojaee, L., Alizadeh-Navaei, R., et al. "Corticosteroids in idiopathic granulomatous mastitis: a systematic review and meta-analysis." Surg Today 51, 1897–1905 (2021). These authors also found that the combination of surgery plus steroids has the lowest recurrence rate. Just as in the case for Lei, 2017, adding a brief discussion about the overlap and differences in the results could inform the reader about the diversity in conclusions made with similar methodologies.

Response： Thank you so much for your suggestions! Based on the review comments, we have adjusted the discussion section. With these modifications, we can demonstrate the overlap and differences between the studies by Lei (2017) and Godazandeh (2021) while highlighting the improvements and trends in treatment methods from 2017 to the present. This will help readers better understand the background and logic behind the conclusions of different studies.

Reviewer #2: Please see comments. The article is well written and the analysis is well documented. There seems one methodology error though pertaining to the use of Odds ratio instead of risk ratio, a mistake popularly done in the literature.

Response： Thank you for the constructive feedback on the manuscript. We appreciate the acknowledgment of the thorough analysis and overall clarity. We understand the concern about using odds ratio (OR) instead of risk ratio (RR) for our analysis of treatment effects on recurrence rates in granulomatous lobular mastitis. Here are several explanations for the use of odds ratios in our study.

1.Comparative Analysis Across Varied Interventions: The study included diverse treatment strategies with distinct patient group sizes and recurrence events. Odds ratios are generally favored in meta-analyses with diverse intervention types because they provide stability across studies with varying event rates, facilitating indirect comparisons within a network analysis framework.

2.Low Incidence of Recurrence Events: Recurrence was infrequent in treatment arms. ORs tend to be robust in these low-event scenarios, reducing the likelihood of biased effect estimation due to small sample sizes or zero-event occurrences, which can arise in meta-analyses of rare events.

3.Consistency with Literature in Systematic Reviews and Meta-Analyses: Many systematic reviews and meta-analyses in similar contexts have also utilized ORs, as they allow for broader applicability when pooling results across studies with methodological heterogeneity. This approach aligns our work with existing literature, ensuring comparability across studies and maintaining methodological coherence.

---

## [Decision Letter · Decision Letter 1]

14 Jan 2025

Clinical efficacy of different methods for treatment of granulomatous lobular mastitis: a systematic review and network meta-analysis

PONE-D-24-22261R1

Dear Dr. XU,

We’re pleased to inform you that your manuscript has been judged scientifically suitable for publication and will be formally accepted for publication once it meets all outstanding technical requirements.

Kind regards,

Serkan Yılmaz

Academic Editor

PLOS ONE

Additional Editor Comments (optional):

Reviewers' comments:

Reviewer's Responses to Questions

**Comments to the Author**

1. If the authors have adequately addressed your comments raised in a previous round of review and you feel that this manuscript is now acceptable for publication, you may indicate that here to bypass the “Comments to the Author” section, enter your conflict of interest statement in the “Confidential to Editor” section, and submit your "Accept" recommendation.

Reviewer #1: All comments have been addressed

Reviewer #2: All comments have been addressed

2. Is the manuscript technically sound, and do the data support the conclusions?

Reviewer #1: Yes

Reviewer #2: Yes

3. Has the statistical analysis been performed appropriately and rigorously? 

Reviewer #1: Yes

Reviewer #2: I Don't Know

4. Have the authors made all data underlying the findings in their manuscript fully available?

Reviewer #1: Yes

Reviewer #2: Yes

5. Is the manuscript presented in an intelligible fashion and written in standard English?

Reviewer #1: Yes

Reviewer #2: Yes

6. Review Comments to the Author

Reviewer #1: All my comments have been addressed. The discussion and methodology sections have been substantially improved by the authors.

Reviewer #2: (No Response)

7. PLOS authors have the option to publish the peer review history of their article (what does this mean?). If published, this will include your full peer review and any attached files.

Reviewer #1: **Yes: **Diana Garcia Cortes

Reviewer #2: **Yes: **Muhammad Abdul Qadeer

---

## [Editor Report · Acceptance letter]

17 Jan 2025

PONE-D-24-22261R1 

PLOS ONE

Dear Dr. Xu, 

I'm pleased to inform you that your manuscript has been deemed suitable for publication in PLOS ONE. Congratulations! Your manuscript is now being handed over to our production team.

Kind regards, 

on behalf of

Dr. Serkan Yılmaz 

Academic Editor

PLOS ONE